# Rildo: Real-World Multicenter Study on the Effectiveness and Safety of Single-Tablet Regimen of Dolutegravir plus Rilpivirine in Treatment-Experienced People Living with HIV

**DOI:** 10.3390/v14122626

**Published:** 2022-11-25

**Authors:** Carmen Hidalgo-Tenorio, David Vinuesa, Coral García-Vallecillos, Leopoldo Muñoz-Medina, Sergio Sequera, Rosario Javier, Miguel Ángel López-Ruz, Svetlana Sadyrbaeva-Dolgova, Juan Pasquau

**Affiliations:** 1Unit of Infectious Diseases, Virgen de las Nieves University Hospital, Biohealth Research Institute, IBS-Granada, 18014 Granada, Spain; 2Unit of Infectious Diseases, San Cecilio University Hospital, Biohealth Research Institute, IBS-Granada, 18012 Granada, Spain

**Keywords:** 2DR, dolutegravir, rilpivirine, treatment-experienced PLHIV

## Abstract

Two-drug regimens (2DRs) are emerging in clinical practice guidelines as treatment option for both naive and treatment-experienced people living with HIV (PLHIV). **Objectives:** To determine the real-life effectiveness of 2DR with 25 mg RPV plus 50 mg DTG in a single-tablet regimen (RPV/DTG_STR_) and its impact on viral and immune status, lipid profile, and inflammatory markers. **Methods:** This observational study included 291 treatment-experienced PLHIV, starting 2DR with RPV/DTG_STR_ between 29 January 2019 and 2 February 2022, who were followed up for at least six months. Participants gave verbal informed consent for the switch in antiretroviral therapy (ART) to RPV/DTG_STR_. **Results:** The mean age of the 291 participants was 51.3 years; 77.7% were male; and 42.9% were in the AIDS stage with a CD4 nadir of 283.5 ± 204.6 cells/uL. The median time since HIV diagnosis was 19.7 years (IQR: 10.6–27). Before 2DR, patients received a median of five ART lines (IQR: 3–7) for 22.2 years (IQR: 14–26), with 34.4% (*n* = 100) receiving a three-drug regimen (3DR), 31.3% (*n* = 91) receiving monotherapy, and 34.4% (*n* = 100) receiving 2DR. The median time on RPV/DTG_STR_ was 14 months (IQR: 9.5–21); 1.4% were lost to the follow-up. Effectiveness was 96.2% by intention-to-treat (ITT) analysis, 97.5% by modified ITT, and 99.3% by per-protocol analysis. Virological failure was observed in 0.69%, blips in 3.5%, and switch to another ART in 1.4%. The mean lipid profile improved, with reductions in TC/HDLc ratio (3.9 ± 0.9 vs. 3.6 ± 0.9; *p* = 0.0001), LDLc (118.3 ± 32.2 mg/dL vs. 106.2 ± 29.8 mg/dL, *p* = 0.0001), TG (130.9 ± 73.9 mg/dL vs. 115.9 ± 68.5 mg/dL, *p* = 0.0001), and CD4/CD8 ratio increase (0.99 ± 0.58 vs. 1.01 ± 0.54; *p* = 0.0001). The cost-effectiveness of 2DR with RPV/DTG_STR_ was similar to that of DTG/3TC and superior to those of BIC/TAF/FTC and DRV/c/TAF/FTC, with higher virological suppression and lower annual costs. **Conclusions:** The switch to RPV plus DTG in STR is a cost-effective, long-lasting, and robust strategy for PLHIV, with a very long experience of treatment, which improves the lipid profile without affecting inflammatory markers.

## 1. Background

Since the introduction of high-effectiveness antiretroviral therapy (ART) in the mid-1990s, the life expectancy of people living with HIV (PLHIV) has progressively increased, approaching that of the general population [1]. For decades, clinical practice guidelines for the standard treatment of naive and treatment-experienced PLHIV has considered only three drug regimens (3DR) as preferential: ART, comprising two nucleoside analogues combined with one integrase inhibitor, a protease inhibitor, and a non-nucleoside analogue [2]. Two-drug regimens (2DRs) have been studied as a means of minimizing short- and long-term antiretroviral (ARV) toxicity in naive or ARV-experienced patients [3]. They include the combination of lamivudine and dolutegravir (DTG), which achieved excellent outcomes in clinical trials in naive [4,5,6] and treatment-experienced patients [7,8,9]. A 2DR with rilpivirine (RPV) plus DTG demonstrated the same effectiveness in treatment-experienced patients as that obtained with three-drug regimens (3DRs) in two clinical trials, SWORD 1 and 2, which found no risk of increased resistance and improvements in renal function and bone metabolism [10]. Although randomized clinical trials are the gold standard for evaluating drug effectiveness, multiple factors can limit the extrapolation of outcomes to the whole population, including the type of patient, diversity of therapeutic response, and adherence to treatment. Accordingly, real-world evidence studies are performed after drug approval in non-selected patients recruited in a routine clinical setting without the restrictions of trials. The aim is to provide in-depth analysis of the “real-life” safety and effectiveness of the drug, based on all available data from a wider group of patients than those enrolled in clinical trials [11]. 

The main objective of this study was to analyze the effectiveness and safety of changing any ARV regimen (1DR, 2DR, or 3DR) to 2DR with RPV (25 mg/QD) plus DTG (50 mg/QD) in a single-tablet regimen (RPV/DTG_STR_) in a cohort of ART-experienced PLHIV. The secondary objectives were to determine the effect of switching on immune and viral status and inflammatory markers and to carry out a pharmacoeconomic study of the change.

## 2. Patients and Methods

### 2.1. Study Design and Setting

A multicenter, observational, and retrospective study was undertaken in PLHIV treated with RPV/DTG_STR_ for at least six months. A 2DR was prescribed by attending physicians after obtaining the informed consent of patients. The study included PLHIV under outpatient treatment at two infectious disease units in the regional health service who had initiated treatment, for any reason, with RPV (25/QD) plus DTG (50 mg/QD) in STR (RPV/DTG_STR_). The study was approved by the ethics committee of Virgen de las Nieves University Hospital of Granada, Spain (#1602-N-22), and data were treated in compliance with Spanish legislation on personal data protection.

### 2.2. Inclusion Criteria

PLHIV treated with any other ART regimen for at least six months who were switched to 2DR with RPV/DTG_STR_ by the attending physician.

### 2.3. Variables/Data Sources

At the baseline visit (V0), data were gathered on age, sex, active co-infections, toxic habits (tobacco and alcohol), mode of acquisition of HIV, weight, date of HIV diagnosis, viral load (VL), CD4 and CD8 lymphocyte counts at diagnosis, CD4 nadir, previous ART lines, reasons for change to RPV/DTG_STR_ (toxicity, intolerance, simplification, optimization, inconvenience, etc.), and analytical data for the month before the change to RPV/DTG_STR_. Data were also gathered at V0 and at the end of the follow-up on: VL, CD4, and CD8 lymphocyte counts and percentages, CD4/CD8 ratio, total cholesterol, HDL, LDL, triglycerides, GOT, GPT, GGT, bilirubin, alkaline phosphatase, interleukin-6, D-dimer, and glomerular filtration (CKD-EPI). 

Treatment effectiveness was determined by calculating the proportion of participants achieving viral suppression (defined by plasma HIV RNA ≤ 50 cop/mL) at the end of follow-up according to intention-to-treat (ITT) [12], modified ITT (mITT) [13], and per-protocol (PP) [14] analyses. 

A blip was defined as VL > 50 copies/mL in one determination and undetectable in the next determination one month later. Virological failure (VF) was defined by VL > 50 HIV-RNA copies/uL at two consecutive determinations in a 6-month period. 

The interpretation of the genotypic resistance test was performed following the algorithms of Stanford University and the Spanish AIDS Research Network (RIS) [15].

It was estimated that at least 235 treatment-experienced PLHIV were needed to measure the effectiveness of 2DR with 95% confidence interval (CI) and 3% precision, assuming a loss to follow-up of 15%. 

Statistical analysis: In the descriptive analysis, frequencies with 95% CI were calculated for qualitative variables, means with standard deviation for normally distributed quantitative variables, and medians with IQR for non-normally distributed quantitative variables. The Kolmogorov–Smirnov test was applied to check the normality of variable distributions. The paired samples Wilcoxon test was used to compare quantitative variables. *p* < 0.05 was considered significant in all tests. SPSS 20.0 was used for statistical analyses.

### 2.4. Pharmacoeconomic Analysis

A pharmacoeconomic analysis was performed, comparing therapy with RPV/DTG vs. BIC/TAF/FTC, DRV/c/TAF/FTC, and DTG/3TC. Data obtained from published research in study populations with similar characteristics [16,17,18] were compared with the effectiveness of RPV/DTG_STR_ by modified ITT analysis (excluding individuals lost to follow-up). Treatment-cost calculations were based on the laboratory price list for February 2022, without VAT and without considering state discounts. Cost-minimization analysis was performed after one year, to determine the savings generated by switching to RPV/DTG. The cost-effectiveness ratio (CER) was also determined by dividing the cost of each treatment by its effectiveness, and the incremental CER (ICER) of RPV/DTG therapy vs. the others was calculated by dividing the difference in total costs (incremental cost) by the increase in effectiveness.

## 3. Results

### 3.1. Study Population

The study included 291 PLHIV with a mean age of 51.3 years; 77.7% (226) were male; and 42.9% had AIDS. The mean CD4 nadir was 283.5 ± 204.6 cells/uL. The median time since HIV diagnosis was 19.7 years (IQR: 10.6–27). Patients had previously received a median of five ART lines (IQR: 3–7) for 15.3 years (IQR: 7.5–2), with 34.4% (*n* = 100) receiving a three-drug regimen (3DR), 31.3% (*n* = 91) receiving monotherapy, and 34.4% (*n* = 100) receiving another 2DR. At baseline (V0), 3.4% (*n* = 10) had VF and 6.2% (*n* = 18) had blips. One patient had a chronic VHB infection (HBS antigen positive) and had been under HBV treatment with entecavir for 10 years due to kidney failure after treatment with 600 mg efavirenz, 200 mg emtricitabine, and 245 mg disoproxil tenofovir in STR (Table 1).

A genotyping resistance test had been performed in 152 (52.2%) of the participants a median of 13.5 years earlier (IQR: 8.3–17.3); among these, 14 (9.25%) had primary and secondary non-nucleoside reverse transcriptase inhibitor (NNRTI)-resistance mutations conferring either resistance (35.7%) or decreased susceptibility to RPV (42.9%) (Table 1).

Reasons for the 2DR switch were simplification in 69.8% (*n* = 203), avoidance of potential long-term toxicity in 26.8% (*n* = 78), and VF in 3.4% (*n* = 10) due to poor ART adherence in a multi-tablet regimen (MTR). The results for the remaining study variables are reported in Table 1. 

### 3.2. DR Effectiveness and Safety

The median time on RPV/DTG_STR_ was 14 months (IQR: 9.5–21); five (1.7%) patients were lost to the follow-up. Effectiveness was recorded in 96.2% (280/291) by ITT, 97.5% (280/287) by modified ITT, and 99.3% (281/283) by PP analyses (Figure 1). 

During the follow-up period, 0.69% of patients (*n* = 2) had VF, 3.5% (*n* = 10) had blips, and 1.4% (*n* = 4) switched from RPV/DTG_STR_ due to drug interactions (*n* = 1), adverse events (diarrhea and depression, *n* = 2), or by decision of the physician (*n* = 1) (Table 2). One of the two patients who failed due to suspension of the treatment did not take even one tablet, and the treatment adherence of the other was inadequate (Table 3). 

### 3.3. Analytical Parameters

During the follow-up, the following were observed: an increase in the CD4/CD8 ratio (0.99 ± 0.58 vs. 1.01 ± 0.54; *p* = 0.0001) and a reduction in the TC/HDLc ratio (3.9 ± 0.9 vs. 3.6 ± 0.9; *p* = 0.0001), LDLc level (118.3 ± 32.2 mg/dL vs. 106.2 ± 29.8 mg/dL, *p* = 0.0001), and TG level (130.9 ± 73.9 mg/dL vs. 115.9 ± 68.5 mg/dL, *p* = 0.0001). No statistically significant changes were found in the IL-6, C-reactive protein, fibrinogen, or D-dimer levels. Table 4 lists the results for the remaining variables. 

### 3.4. Pharmacoeconomic Analysis

The annual cost per patient was EUR 5538.96 for RPV/DTG_STR_, EUR 6012.36 for BIC/TAF/FTC, EUR 6094.56 for DRV/c/FTC/TAF, and EUR 4851.24 for DTG/3TC. Switching to 2DR with RPV/DTG_STR_ would, therefore, yield a potential annual saving per patient of EUR 473.40 vs. BIC/FTC/TAF and EUR 555.60 vs. DRV/c/FTC/TAF. The annual cost per patient would increase by EUR 687.72 in comparison to DTG/3TC (Table 5a).

The virological effectiveness ratio was 97.5% for RPV/DTG_STR_, 94.8% for BIC/FTC/TAF, 90.7% for DRV/c/FTC/TAF, and 87.9% for DTG/3TC, and the CER values were around 57, 63, 67, and 55, respectively. The cost-effectiveness of RPV/DTG_STR_ to treat HIV is, therefore, comparable to that of the other ARTs. The ICER of RPV/DTG_STR_ was −175 vs. BIC/FTC/TAF, −82 vs. DRV/c/FTC/TAF, and 87 vs. DTG/3TC (Table 5b). These findings place RPV/DTG_STR_ as a cost-effective option with respect to the other therapies, with improved virological suppression and a slightly lower annual cost (except for the other 2DR with DTG/3TC).

## 4. Discussion

Most patients in this real-world study on the use of 2DR with RPV/DTG_STR_ in ART-experienced PLHIV were aged over 50 years, and around half of them were in the AIDS stage. They had a very long history of ART (median of 22.1 years), with a median of five previous lines. They were switched from different ARV regimens and even included a small percentage of non-suppressed patients. Mutations conferring NNRTI resistance had been detected in 5% over 13 years earlier, and previous exposure to integrase standard transfer inhibitors (INSTIs) was recorded in 30%, but none had resistance mutations to DTG. The majority of participants in SWORD 1 and 2 clinical trials were aged <50 years, while around 11% were in the AIDS stage, and the median time with ART was four years, with a maximum of two previous lines; in addition, they were all suppressed before starting the 2DR, <10% had been exposed to DTG, and none had resistance mutations to NNRTI [19]. 

In the present study, the reason for switching was simplifying the regimen in more than two-thirds of cases, avoiding long-term toxicity in more than one-quarter, and failing to adequately adhere to their MTR in one-tenth; finally, clinicians even trusted in this regimen in one out of ten patients in failure due to poor adherence to treatments in MTR. Owing to the increased survival and life expectancy of PLHIV achieved by ART, it has been estimated that patients will consume ARVs for a mean of 39.1 years, receiving 57,086 ARVs for 3DR with a booster, 42,815 for 3DR without a booster, and 28,543 for 2DR [20]. The potential benefits of a paradigm change from 3DR to 2 DR regimens include a decrease in long-term adverse effects and toxicities, a reduced possibility of interactions, the preservation of ARVs not included in the regimens, improvements in quality of life, and a reduction in treatment costs [21]. 

Only four patients were lost to the follow-up of more than 12 months, 2DR with RPV/DTG_STR_ achieved an effectiveness of 95% by ITT analysis and close to 100% by PP analysis in this cohort of ART-experienced PLHIV, and there was no failure in patients with RPV resistance mutations. These effectiveness data are highly similar to those obtained in registered clinical trials [19]. Despite the likelihood of VF due to archived RPV resistance mutations, a high rate of exposure to INSTIs, and a number of previous ARV lines, the effectiveness of treatment was comparable to that reported by the SWORD trials [18], indicating the robustness and effectiveness of this 2DR in the real world. The maintenance of untransmittable status is of major importance not only for the “treatment as prevention” strategy but also to improve the quality of life of PLHIV, reducing self-stigma. 

Only two patients had adverse effects attributable to 2DR with RPV/DTG_STR_ (diarrhea and depression, respectively). In addition, the treatment was only suspended in two patients, to avoid drug interactions and for the convenience of the patient as decided by the physician. Finally, VF was observed in two patients, which was attributed to poor treatment adherence, with neither having mutations in the historical genotype, demonstrating the robustness of the treatment and/or that strains with mutations were a very small minority and were completely suppressed after a long time with a previous effective ART. These findings are in agreement with a trial of this 2DR that showed the sustained virological suppression of around 90% after a three-year follow-up, with low VF rates and a well-tolerated safety profile [22].

The treatment improved cholesterol and triglyceride levels and the CT/HDL ratio, as also found in clinical trials [19] and previous real-life studies [23,24]. Hence, this 2DR is of interest as an option for older PLHIV with risk factors and/or cardiovascular disease. In comparison to the present investigation, the aforementioned studies [23,24] had a smaller sample size, a shorter follow-up of 24 weeks [23], and a different multi-tablet 2DR regimen; furthermore, all PLHIVs were virologically suppressed for at least six months before the switch to 2DR, which was a second-line treatment in the majority of cases. 

An improvement in immunological recovery, with an increased CD4/CD8 ratio, was recorded at one year of treatment with RPV/DTG_STR_. A recent retrospective, multicenter Spanish study of the switch from 3DR to 2DR with RPV/DTG in virologically suppressed PLHIV also found an increase in CD4 and a decrease in CD8 at 48 weeks [25]. Bone metabolism was not analyzed in the present study, but an increase in bone density and decrease in turnover markers was reported in patients changing from a 3DR ART with tenofovir disoproxil fumarate (TDF) to 2DR with RPV/DTG_STR_ [26]. No changes in inflammatory markers (PCR, IL-6, fibrinogen, and D-dimer) were observed during the present follow-up period, and the SWORD trials also found no differences in inflammatory and atherogenic markers at 48 or 148 weeks of treatment with RPV/DTG_STR_ [27].

The switch to RPV/DTG_STR_ in stable and mostly virologically suppressed ART-experienced PLHIV proved to be a cost-effective alternative to 3DR ART, offering higher effectiveness at a lower cost in comparison to BIC/FTC/TAF and DRVc/FTC/TAF. These results are in line with previous findings of cost saving with DTG/RPV vs. RPV/FTC/TDF, DTG/ABC/3TC, and EVG/c/FTC/TAF and the delivery of comparable efficacy with reduced cumulative drug exposure [28]. Finally, the cost-effectiveness was highly similar to that of 2DR with DTG/3TC, indicating that 2DR ART with RPV/DTG in STR can stand alongside DTG/3TC as a preferential strategy in treatment-experienced PLHIV. However, these findings do not rule out the other therapies as efficient options. 

The main study limitation is its retrospective design. Strengths include the large sample size of this multicenter study. In addition, it is the first to study the switch to this 2DR in PLHIV with long ART experience and no need for treatment suppression for >6 months, including individuals with archived RPV resistance mutations (none failed) and a high rate of exposure to integrase inhibitors. All these characteristics make this a unique study of major interest. 

In conclusion, 2DR with RPV plus DTG in STR is an effective, safe, long-lasting, robust strategy for patients with prolonged ART experience, improving their lipid profile without affecting inflammatory markers; it is cost-effective compared to the other treatments.

## Figures and Tables

**Figure 1 viruses-14-02626-f001:**
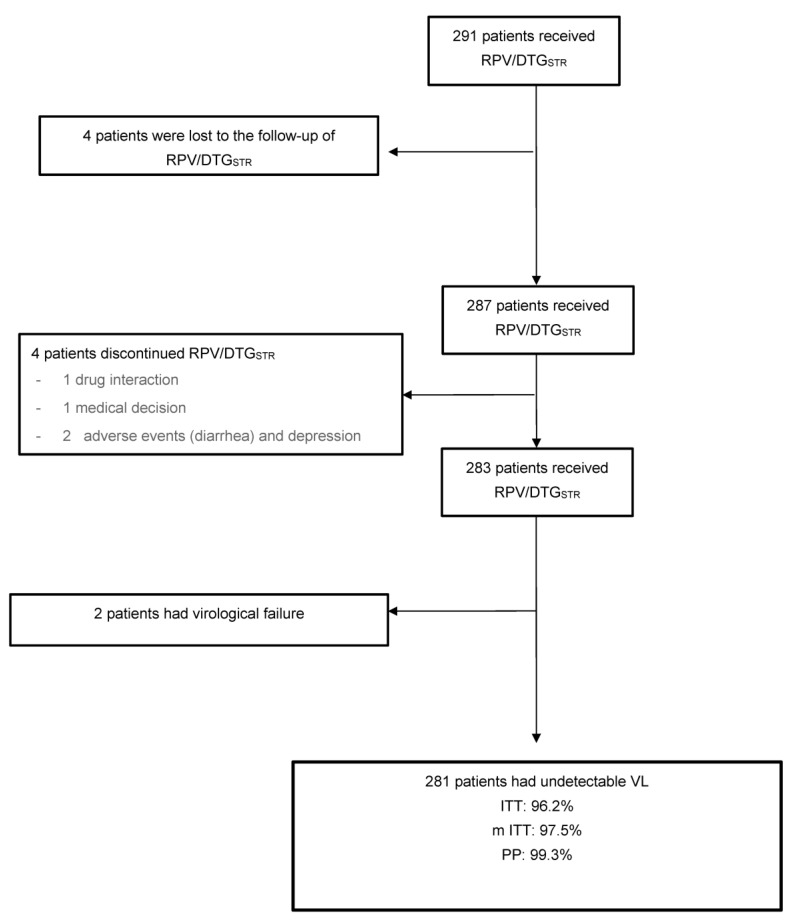
Flowchart of patients with rilpivirine plus dolutegravir (STR).

**Table 1 viruses-14-02626-t001:** General description of 2DR population.

Variable	N = 291
Age (years), mean (± SD)	51.3 (±11.4)
Male, *n* (%)	226 (77.7)
Time from HIV diagnosis (year), median (IQR)	19.7 (10.6–27)
CD4 nadir, mean (± SD)	283.5 ± 204.6
Baseline VL < 50 copies/mL, *n* (%), 95% CI	287 (98.6) (97.1–99.9)
Viral load of HIV, log10, median (IQR)	0 (0–0)
Baseline CD4, (cell/uL), mean (± SD)	785.4 (±355.7)
Baseline CD4/CD8 ratio, mean (± SD)	0.99 (±0.58)
History of AIDS (A3, B3, C), *n* (%), 95% CI	12 5(42.9) (37.9–49.7)
HBs antigen positive, *n* (%) HB core antibody positive, *n* (%), 95% CI IgG HCV-positive (cured), *n* (%), 95% CI History of infectious transmission diseases, *n* (%) Smoking, *n* (%), CI 95%	1 (0.3) 8 (2.7) (1.2–5.4) 67 (23) (18.8–29) 35 (12) 155 (53.3) (44.8–56.7)
Risk factor for HIV infection, *n* (%) - Heterosexual - MSM - Ex-IVDU - Other	54 (18.6) 184 (63.2) 48 (16.5) 2 (2.2)
Previous ART lines, median (IQR)	5 (3–7)
Time on ART (years), median (IQR)	15.3 (7.5–22.1)
Previous ART regimen, *n* (%) **3DR** INSTI NNRTI PI **2DR** INSTI + NNRTI Lamivudine + PI PI + NNRTI **1DR** DRV/r or cob LOP/r	100 (34.4) 53 (53) 20 (20) 27 (27) 91 (31.3) 40 (43.9) 26 (28.6) 25(27.4) 100 (34.4) 98 (98) 2 (2)
History of ARV with INSTI, *n* (%) History of ARV with NNRTI, *n* (%)	87 (29.9) 111 (38.1)
Time (years) since genotype resistance test, median (IQR)	13.5 (8.3–17.3)
Resistances to a NNRTI by patient	**Resistance to RPV** Y 181C, V179I Y181C, G190A, M184V M 184v, Y181C, V179I K103N, Y181C, V 106M K103N, V108I, M184I **Decreased susceptibility to RPV** K103N, G190A, M184V (2 patients) M184V, V106A, V 179D (2 patients) K103N, V108I, Y181C K101/E, G190A **Sensible to RPV** K103N, E138A K103N, M184V, P225HG106A, M184V
Median time (months) of RPV/DTG_STR_ administration (IQR)	14 (9.5–21)
Reason for change to RPV/DTG_STR_, *n* (%)	
Simplification/optimization Avoiding long-term drug toxicity Virological failure due to poor adherence to MTR	203 (69.8) 78 (26.8) 10 (3.4)

**MSM**: men who have sex with men; **Ex-IVDU**: ex-intravenous drug user; **RPV**: rilpivirine; **DTG**: dolutegravir; **_STR_**: single-tablet regimen; **3DR**: three-drug regimen; **2DR**: two-drug regimen; **1DR**: monotherapy; **NNRTI**: non-nucleoside reverse transcriptase inhibitor; **INSTI**: integrase standard transfer inhibitor; **MTR**: multi-tablet regimen; **PI**: protease inhibitor.

**Table 2 viruses-14-02626-t002:** Outcomes of RILDO study.

Effectiveness	N = 291
Intention-to-treat (ITT) analysis, (%) *n*/N * Modified ITT (mITT) analysis, (%) *n*/N ** Per-protocol (PP) analysis, (%) *n*/N ***	96.2 (280/291) 97.5 (280/287) 99.3 (281/283)
	*n* = 287
Blips, *n* (%)	10 (3.5)
Virological failure, *n* (%)	2 (0.69)
Adverse effects, *n* (%)	2 (0.69)

*n* = number of patients with undetectable viral load at end of follow-up. N * = number of patients in the study. N ** = number of patients in the cohort, excluding losses to follow-up. N *** = number of patients included, excluding those who discontinued treatment due to adverse effects, to avoid drug interactions, or by physician decision.

**Table 3 viruses-14-02626-t003:** Virological failure.

PLHIV	Age Years	S	Years Since HIV Diagnosis Stage	ART (Total Time, Previous ART and Time) Previous Lines	Baseline VL	VL in VF	** RS	Month VF	Observations
1	59	M	28.7 and B3	27 years in ART DRV/cob (49 months) 8 lines	<50 cop/mL	173,000 cop/mL	Not available	3 months	Abandoned ART and follow-up
2	57	F	33.3 and B2	26 years in ART EVG/c/TAF/FTC (27 months) 14 lines	25,700 cop/mL	122,200 cop/mL	Not available	19 months	Null adherence

M: male; F: female; ART: antiretroviral therapy; DRV/cob: darunavir/cobicistat; EVG/C/FTC/TAF: elvitegravir/cobicistat/emtricitabine/tenofovir alafenamide; VL: HIV viral load; VF: virological failure; ** RS: genotypic resistance test.

**Table 4 viruses-14-02626-t004:** Analytical changes between baseline visit and last visit.

	Baseline	Last Visit	*p* Value
CD4 (cells/uL), mean ± SD	785.4 ± 355.7	772.7 ± 342.3	0.43
CD4/CD8 ratio, mean ± SD	0.99 ± 0.58	1.01 ± 0.54	0.013
Creatinine clearance mL/h, mean ± SD	117.5 ± 458.3	122.8 ± 592.3	0.001
Total cholesterol (mg/dL), mean ± SD	190.6 ± 39.4	176.8 ± 35.5	0.0001
HDL cholesterol (mg/dL), mean ± SD	52.1 ± 15.4	54.2 ± 15.1	0.482
LDL cholesterol (mg/dL), mean ± SD	118.9 ± 32.2	106.2 ± 29.8	0.0001
TC/HDL ratio, mean ± SD	3.9 ± 0.9	3.6 ± 0.9	0.0001
Triglycerides (mg/dL), mean ± SD	130.9 ± 73.9	115.3 ± 68.5	0.0001
Bilirubin (mg/dL), mean ± SD	0.7 ± 0.4	0.7 ± 0.3	
GPT (UI/dL), mean ± SD	26.8 ± 21.9	29.5 ± 25.1	0.0001
GGT (UI/dL), mean ± SD	38.9 ± 63.8	34.9 ± 56.2	0.794
FA (UI/dL), mean ± SD	77.8 ± 39.6	73.4 ± 32.5	0.0001
Interleukin-6 (pg/mL), mean ± SD	3.9 ± 5.8	4.1 ± 0.6	0.851
D-dimer (UEF/mL), mean ± SD	0.4 ± 0.4	0.4 ± 0.6	0.251
Fibrinogen, mg/dL, mean ± SD	283.9 ± 92.3	273.5 ± 116.8	0.054
PCR, mg/dL	3.3 ± 4.2	3.1 ± 4.5	0.093

**Table 5 viruses-14-02626-t005:** (a) Cost minimization among the different ARTs. (b) Cost-effectiveness table.

**(a) Cost Minimization among the Different ARTs**				
**ARTs Compared**		**Differential Cost Per Year**		
RPV/DTG_STR_ vs. BIC/TAF/FTC		EUR −473.40		
RPV/DTG_STR_ vs. DRV/c/TAF/FTC		EUR −555.60		
RPV/DTG_STR_ vs. DTG/3TC		EUR 687.72		
**(b) Cost-Effectiveness Table**				
**ART**	**Effectiveness**	**Cost (€/Year)**	**CER**	**ICER**
DTG/3TC	87.9%	EUR 4851	55	87
BIC/FTC/TAF	94.8.0%	EUR 6012	63	−175
DRV/c/FTC/TAF	90.7%	EUR 6095	67	−82
RPV/DTG_STR_	97.5%	EUR 5539	57	-

CER cost-effectiveness ratio; ICER: incremental cost-effectiveness ratio. Bictegravir/emtricitabine/tenofovir alafenamide (BIC/FTC/TAF); darunavir/cobicistat/emtricitabine/tenofovir alafenamide (DRV/c/FTC/TAF).

## Data Availability

The authors confirm the accuracy of the data provided for the study as well as its availability.

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
