# Peer review of "Rildo: Real-World Multicenter Study on the Effectiveness and Safety of Single-Tablet Regimen of Dolutegravir plus Rilpivirine in Treatment-Experienced People Living with HIV"

_viruses, 2022, doi:10.3390/v14122626_

Round 1

Reviewer 1 Report

Review-20221026

This is a retrospective observational study that looked at dolutegravir plus rilpivirine as a single tablet regimen safety and efficacy in a treatment-experienced population of mostly virologically suppressed patients.

Overall, the results demonstrate the efficacy of this drug combination in maintaining viral suppression. It is a well-written and timely paper on a two-drug regimen for HIV treatment. However, I am unconvinced by some of the statistical significance of some of the differences that are reported, including lipid health. This serious concern needs to be addressed, and the conclusions and abstract must be modified accordingly. In addition, the cost-effectiveness study does not support scrutiny as the comparison is made on cohorts very different from the one described here. These differences in population make the cost-effectiveness results exceedingly biased, and they are not supported by evidence. The cost-effectiveness results must be removed from the manuscript.

My major and minor comments are explained below.

Major comments:

The statistical tests used to compare baseline to final visit outcomes must be mentioned in the materials and methods. Was it a parametric or non-parametric test? Was a test for normality used? (These should also be reported in the Methods). A simple t-test on the CD4/CD8 ratio, as an example, shows the absence of significance between 0.99 and 1.01 with the indicated error margins. It is, in fact, unlikely that these numbers should be statistically different by any rational statistical means. It is not unsurprising that most virologically suppressed individuals who have been using an effective regimen for several years could have already corrected their CD4 counts. The same applies to creatinine clearance, GPT (both not statistically significant by simple t-test, inferred from presented data), and potentially to several other parameters that are indicated as statistically significant by the authors (in Table 4). The authors must re-perform their statistical analysis and revise their conclusion when needed.

The authors propose a cost-effectiveness indirect comparison with several other antiretroviral regimens. However, I am worried that the comparisons are not properly controlled. The difference is most particularly shocking for BIC/F/TAF, for which the authors attribute an 85% mITT suppression rate, based on one publication (Ambrosioni, JAC, 2022). However, the BIC publication had a cohort different than the one who received dolutegravir plus rilpivirine. Most notably, only 82% of treatment-experienced individuals were suppressed at baseline vs. 98.6% in the current study. The cohort was younger, had a lower CD4 count, and had a shorter (about ½) time since diagnosis. The same objections stand for the apparent high efficacy of DRV/c/TAF/FTC that can be attributed to the fact that participants in the EMERALD clinical trial (that the authors used for comparison) were 100% virologically suppressed (only one low-level viremia measure was allowed in the past year for inclusion). Since the cohorts are different, the cost-effectiveness analysis must not be included in the paper.

The only way this analysis could be done correctly would be for the authors to get data from the various clinical trials used as a comparator and pool this data to create groups that match clinical characteristics (and thus can be compared), then perform a cost study.

One of the major concerns regarding switching to 2DR regimens without tenofovir is HBV reactivation. The authors must indicate the HBV status of the participants. I assume that all were HBV-negative, but this information must be provided. This information is vital given the current clinical context surrounding the diagnosis and treatment of this other virus.

The authors report on lipid outcomes. Were these parameters not reported in the two SWORD clinical trials? If they were, the authors must include these results in their introduction. If they were not, the authors should highlight how their current work is unique in this regard (in the introduction).

Minor comments:

The fact that this was an observational study should be mentioned in the abstract.

Line 62: “inflammatory markers” as a secondary outcome; however, the authors only report liver, bone, and lipid clinical parameters in the Methods (line 83). From the illustrations, I can see that IL6 and D-dimer were collected. The authors must amend the Materials and Methods section to list all clinical parameters that were collected, ideally categorizing them by system (liver, bone, inflammation, immunological).

Median time since diagnosis was 18.5 (abstract) or 19.7 (line 116)?

Line 123: among the 14 patients, the authors must provide the percentage of those who were predicted by an algorithm to be fully resistant against RPV vs. those who were with decreased susceptibility. I mean, how many of those 14 patients were fully resistant? This information is crucial for the interpretation of the data and guidance to our colleagues

In table 1, the authors indicate 98.6% viral suppression at baseline but a 4.56 log 10 median VL. I assume that the data indicated here is the VL at treatment initiation. This should be clarified in the table.

Throughout: the usual terminology is PLWHIV, not “PLHIV”. I encourage the authors to use common terminology in their manuscript.

Typo in the abstract line 25 (two times “97.5%”)

Line 45: “excellent outcomes” is a non-scientific qualificative, it would be better to give a range of viral suppression rates at a specific time (for example: “xx to xx% viral suppression after xx weeks in naïve…”

Line 57: “patients than those enrolled in clinical trials”

Line 68, “regional health service”: please add which region that is, in which country. I can infer this information based on the affiliations of the authors, but it would be better to have this information in the text. Line 70, “the hospital”: which hospital, location, city, country. If you have an ethics approval number, you should include it here.

Line 76, “toxic habits” sounds a bit harsh; the authors should use “alcohol and tobacco consumption” (plus illegal drug use if this information was also collected). Was the use of supplements and vitamins also collected? If yes, it could be added here. Line 77, same for “risky practices”: I don’t know what that means (is riding a motorbike considered a risky practice?), but I assume that questions about sexual behaviors were asked. I disagree with the importance of this question in regard to the clinical parameters and outcomes that are reported here. Thus, I suggest the authors clarify respectfully or simply remove “risky practices” from their text. The suspected mode of acquisition as reported in the Table is fine.

Line 91: “it was as estimated”: remove “as”

Table 1 “cell” instead of “cel”.

In table 1, what is “HCB co-infection”? Is it chronic hepatitis B infection?

Line 286: text from recommendations to authors to be edited or removed.

In figure 1, Spanish language in the lower box. Please translate.

Table 5a: “differential costs” add “per year”

Author Response

Reviewer 1:

This is a retrospective observational study that looked at dolutegravir plus rilpivirine as a single tablet regimen safety and efficacy in a treatment-experienced population of mostly virologically suppressed patients.

Overall, the results demonstrate the efficacy of this drug combination in maintaining viral suppression. It is a well-written and timely paper on a two-drug regimen for HIV treatment. However, I am unconvinced by some of the statistical significance of some of the differences that are reported, including lipid health. This serious concern needs to be addressed, and the conclusions and abstract must be modified accordingly. In addition, the cost-effectiveness study does not support scrutiny as the comparison is made on cohorts very different from the one described here. These differences in population make the cost-effectiveness results exceedingly biased, and they are not supported by evidence. The cost-effectiveness results must be removed from the manuscript.

My major and minor comments are explained below.

Major comments:

The statistical tests used to compare baseline to final visit outcomes must be mentioned in the materials and methods. Was it a parametric or non-parametric test? Was a test for normality used? (These should also be reported in the Methods).

This part has been rewritten to address these questions, as follows. “The Student’s t-test was used for the descriptive analysis of quantitative variables with normal distribution and Mann-Whitney U test for those with non-normal distribution, using the Kolmogorov-Simirnov test to determine the normality of variable distribution. The Wilcoxon test was used to compare quantitative variables. p<0.05 was considered significant in all tests. SPSS20.0 was used for statistical analyses”.

A simple t-test on the CD4/CD8 ratio, as an example, shows the absence of significance between 0.99 and 1.01 with the indicated error margins. It is, in fact, unlikely that these numbers should be statistically different by any rational statistical means. It is not unsurprising that most virologically suppressed individuals who have been using an effective regimen for several years could have already corrected their CD4 counts. The same applies to creatinine clearance, GPT (both not statistically significant by simple t-test, inferred from presented data), and potentially to several other parameters that are indicated as statistically significant by the authors (in Table 4). The authors must re-perform their statistical analysis and revise their conclusion when needed. These parameters were compared with the Wilcoxon test and not with the Student’s t-test, as now clarified (see above).

The authors propose a cost-effectiveness indirect comparison with several other antiretroviral regimens. However, I am worried that the comparisons are not properly controlled. The difference is most particularly shocking for BIC/F/TAF, for which the authors attribute an 85% mITT suppression rate, based on one publication (Ambrosioni, JAC, 2022). However, the BIC publication had a cohort different than the one who received dolutegravir plus rilpivirine. Most notably, only 82% of treatment-experienced individuals were suppressed at baseline vs. 98.6% in the current study. The cohort was younger, had a lower CD4 count, and had a shorter (about ½) time since diagnosis. The same objections stand for the apparent high efficacy of DRV/c/TAF/FTC that can be attributed to the fact that participants in the EMERALD clinical trial (that the authors used for comparison) were 100% virologically suppressed (only one low-level viremia measure was allowed in the past year for inclusion). Since the cohorts are different, the cost-effectiveness analysis must not be included in the paper.

The only way this analysis could be done correctly would be for the authors to get data from the various clinical trials used as a comparator and pool this data to create groups that match clinical characteristics (and thus can be compared), then perform a cost study.

We are grateful for these observations on the cost-effectiveness study. It should be borne in mind that few real-life studies have evaluated these therapies in pretreated patients with comparable characteristics, based on a similar sample size and follow-up time to the present investigation, and we have selected those considered most relevant. Following this reviewer’s comment, however, the article previously cited on BIC/F/TAFhas been replaced with the following more recent study, which is more comparable to our investigation. (Mazzitelli, M.; Trunfio, M.; Putaggio, C.; Sasset, L.; Leoni, D.; Lo Menzo, S.; Mengato, D.; Cattelan, A.M. Viro-Immunological, Clinical Outcomes and Costs of Switching to BIC/TAF/FTC in a Cohort of People Living with HIV: A 48-Week Prospective Analysis. Biomedicines 2022, 10, 1823. https://doi.org/10.3390/biomedicines10081823)

We are aware that the article on DRV/c/F/TAF reports a clinical trial that is therefore strongly controlled (e.g., virologically suppression was an inclusion criterion). However, we were unable to trace any other real-life study (Real World Evidence) with similar power or data for comparison with our findings

.

The following data from our pharmacoeconomic study are now reported:

ART

Effectiveness

Cost (€/year)

CER

ICER

DTG+3TC 

87.9%

4,851 €

55

87

BIC/TAF/FTC

94.8%

6,012 €

63

-175

DRV/c/TAF/FTC

90.7%

6,095 €

67

-82

RPV/DTG 

97.5%

5,539 €

57

-

One of the major concerns regarding switching to 2DR regimens without tenofovir is HBV reactivation. The authors must indicate the HBV status of the participants. I assume that all were HBV-negative, but this information must be provided. This information is vital given the current clinical context surrounding the diagnosis and treatment of this other virus.

We report on the HBV status of patients in Table 1, showing that

eight patients were Ab core HBV positive, Ag sHBV negative, and Ab sHBV positive. One patient was Ag sHBV positive and had been under HBV treatment with entecavir for 10 years due to kidney failure after treatment with TDF/FTC/efavirenz in STR .

The authors report on lipid outcomes. Were these parameters not reported in the two SWORD clinical trials? If they were, the authors must include these results in their introduction. If they were not, the authors should highlight how their current work is unique in this regard (in the introduction).

The novelty of our study is not the measurement of these parameters but rather the type of patient. In Sword trials, the switch was in patients who had received a maximum of 2 antiretroviral treatment lines, had no previous experience with integrase inhibitors, had no mutations of resistances to the treatment regimen of DTG/RPV, and had been diagnosed recently. By contrast, RILDO is a real-world, observational study in “very experienced -antiretroviral HIV patients”, with 5 lines of previous ART and experience in integrase inhibitors in >30% and in non-analogs, with rilpivirine-resistance mutations.

Minor comments:

The fact that this was an observational study should be mentioned in the abstract. This has been done.

Line 62: “inflammatory markers” as a secondary outcome; however, the authors only report liver, bone, and lipid clinical parameters in the Methods (line 83). From the illustrations, I can see that IL6 and D-dimer were collected. The authors must amend the Materials and Methods section to list all clinical parameters that were collected, ideally categorizing them by system (liver, bone, inflammation, immunological). We have rewritten the second objective accordingly: “to measure the effect of the change (switch) on immunological status, inflammatory markers, and lipid, hepatic, and renal profile); besides performing a pharmacoeconomic study of the change”

Median time since diagnosis was 18.5 (abstract) or 19.7 (line 116)? This typographic error in the Abstract has been corrected (to 19.7).

Line 123: among the 14 patients, the authors must provide the percentage of those who were predicted by an algorithm to be fully resistant against RPV vs. those who were with decreased susceptibility. I mean, how many of those 14 patients were fully resistant? This information is crucial for the interpretation of the data and guidance to our colleagues.

This information has been included in the revised text:  ”reverse transcriptase inhibitor (NNRTI), resistance mutations conferring either resistance (35.7%) or decreased susceptibility to RPV (64.2%) (Table 1)”. The interpretation of the genotypic resistance test was performed following the algorithms of Stanford University and the Spanish AIDS Research Network (RIS) (15). 15- Imaz A, Garcia F, di Yacovo S, Llibre JM. Perfil de resistencias a rilpivirina. Enferm. Infecc. Microbiol. Clin. 2013; 31: 36-43.

In table 1, the authors indicate 98.6% viral suppression at baseline but a 4.56 log 10 median VL. I assume that the data indicated here is the VL at treatment initiation. This should be clarified in the table. Table 1 shows that 287 of the 291 patients who began DTG/RPV were undetectable. Baseline VL< 50 cop refers to the VL before beginning any ARV; the VL before initiating DTG/RPV in STR was 0 (IQR, 0-0). This has been clarified.

Throughout: the usual terminology is PLWHIV, not “PLHIV”. I encourage the authors to use common terminology in their manuscript.According to our review of the literature, both terms are widely employed. For instance, PLHIV is used in the following articles: Haoyi W, et al. “The likelihood of severe COVID-19 outcomes among PLHIV with various comorbidities: a comparative frequentist and Bayesian meta-analysis approach”. J Int AIDS. Soc 2021; 24. Reijer J. Employment trajectories of PLHIV on ART in Lusaka, Zambia: a short report. AIDS Care 2021; 33; Jespersen NA et al. The burden of non-communicable diseases and mortality in people living with HIV (PLHIV) in the pre-, early- and late-HAART era. HIV Med 2021; 22:

Nevertheless, we are happy to make this change if preferred by the editorial team.

Typo in the abstract line 25 (two times “97.5%”). This has been corrected.

Line 45: “excellent outcomes” is a non-scientific qualificative, it would be better to give a range of viral suppression rates at a specific time (for example: “xx to xx% viral suppression after xx weeks in naïve…”.

This has been done, as follows: “They included the combination of lamivudine and dolutegravir (DTG), which achieved 86.3-93% virological suppression in clinical trials in naïve patients (4, 5, 6), and 82.4-99.7% undetectable viral load in treatment-experienced patients (7, 8, 9).

Line 57: “patients than those enrolled in clinical trials”. This change has been made.

Line 68, “regional health service”: please add which region that is, in which country. I can infer this information based on the affiliations of the authors, but it would be better to have this information in the text. Line 70, “the hospital”: which hospital, location, city, country. If you have an ethics approval number, you should include it here. This information is now given: Virgen de las Nieves University Hospital of Granada, Spain (#1602-N-22)

Line 76, “toxic habits” sounds a bit harsh; the authors should use “alcohol and tobacco consumption” (plus illegal drug use if this information was also collected). This change has been made.

Was the use of supplements and vitamins also collected? If yes, it could be added here. These data were not gathered.

Line 77, same for “risky practices”: I don’t know what that means (is riding a motorbike considered a risky practice?), but I assume that questions about sexual behaviors were asked. I disagree with the importance of this question in regard to the clinical parameters and outcomes that are reported here. Thus, I suggest the authors clarify respectfully or simply remove “risky practices” from their text. The suspected mode of acquisition as reported in the Table is fine. We have replaced this term with “”mode of HIV acquisition”.

Line 91: “it was as estimated”: remove “as”. This change has been made.

Table 1 “cell” instead of “cel”.      This change has been made.

In table 1, what is “HCB co-infection”? Is it chronic hepatitis B infection? This is now written in full

Line 286: text from recommendations to authors to be edited or removed This has been removed.

In figure 1, Spanish language in the lower box. Please translate. This error has been amended

Table 5a: “differential costs” add “per year” This has been done.

Reviewer 2 Report

The ms is well-written and the presented data from the RILDO study (Real-World Multicenter Study on the Effectiveness and 2 Safety of Single-Tablet Regimen of Dolutegravir Plus Rilpi-3 virine in Treatment-Experienced People Living with HIV) are well described and of interest for clinicians interested to know the real-life effectiveness of a commercial available 2DR ART scheme

just a minor comment re table 1; it would be more informative to add infos re previously experienced ART schemes (even in categories)

Author Response

Reviewer 2:

The ms is well-written and the presented data from the RILDO study (Real-World Multicenter Study on the Effectiveness and 2 Safety of Single-Tablet Regimen of Dolutegravir Plus Rilpi-3 virine in Treatment-Experienced People Living with HIV) are well described and of interest for clinicians interested to know the real-life effectiveness of a commercial available 2DR ART scheme

just a minor comment re table 1; it would be more informative to add infos re previously experienced ART schemes (even in categories).

We are grateful for these positive comments. As suggested, we have added data on previous ART regimens in Table 1.

Reviewer 3 Report

Randomized trials have shown effectiveness and safety of the 2 drug regimen RPV/DTG. Real world evidence (RWE) in a more diverse population and with longer follow-up is needed. A retrospective cohort was created of treatment experienced patients, from two clinic sites in Granada, Spain, who switched to RPV/DTG for any reason.  While a worthy goal, 291 patients followed for a median of 14 months is quite limited for RWE. Most RWE studies are much larger and more comprehensive. The analysis and reporting are lacking in several areas.

 Viral suppression was defined as <50.  Blips and virological failure were defined as >50.  Where does VL = 50 fall?

 Statistical analysis is inadequately described.  There is no indication of what test was used for the results reported in Table 4, comparing means at baseline and last visit and including a p-value. Because these are pre-post results on the same people a paired sample test is needed.

 Table 1 reports that 98.6% were virally suppressed at baseline.  Viral load is reported as median 4.56.  This doesn’t make sense.  Is this the viral load for the 1.4% that were not suppressed?

No mention is made of missing data.  I find it hard to believe that all patients had complete information for all the labs listed in Table 4, especially in a retrospective study.  I would be surprised if IL-6 and d-dimer for example are considered routine tests run on everyone.

Author Response

Reviewer 3:

Randomized trials have shown effectiveness and safety of the 2 drug regimen RPV/DTG. Real world evidence (RWE) in a more diverse population and with longer follow-up is needed. A retrospective cohort was created of treatment experienced patients, from two clinic sites in Granada, Spain, who switched to RPV/DTG for any reason.  While a worthy goal, 291 patients followed for a median of 14 months is quite limited for RWE. Most RWE studies are much larger and more comprehensive. The analysis and reporting are lacking in several areas. Viral suppression was defined as <50.  Blips and virological failure were defined as >50.  Where does VL = 50 fall?

R: This was a typographical error; virological suppression was defined as ≤ 50 cop/mL.

 Statistical analysis is inadequately described.  There is no indication of what test was used for the results reported in Table 4, comparing means at baseline and last visit and including a p-value. Because these are pre-post results on the same people a paired sample test is needed.

R: This section has been rewritten and now describes all tests used:

The Student’s t-test was used for the descriptive analysis of quantitative variables with normal distribution and Mann-Whitney U test for those with non-normal distribution, using the Kolmogorov-Simirnov test to determine the normality of variable distribution. The Wilcoxon test was used to compare quantitative variables. p<0.05 was considered significant in all tests. SPSS20.0 was used for statistical analyses”.

 Table 1 reports that 98.6% were virally suppressed at baseline.  Viral load is reported as median 4.56.  This doesn’t make sense.  Is this the viral load for the 1.4% that were not suppressed?  Table 1 shows that 287 of the 291 patients who began DTG/RPV were undetectable. ”Baseline VL< 50 cop” refers to the VL before starting any ARV; the VL before initiating DTG/RPV in STR was 0 (IQR, 0-0).

R: This has been clarified.

No mention is made of missing data.  I find it hard to believe that all patients had complete information for all the labs listed in Table 4, especially in a retrospective study.  I would be surprised if IL-6 and d-dimer for example are considered routine tests run on everyone.

R: The routine care of HIV patients in the two participating hospitals involves a systematic analytical study that includes lipid, renal, and Ca-P metabolism profiles and inflammatory markers. Inclusion criteria for the study required at least 6 months of follow-up with the 2DR and two analytical determinations, comparing baseline and final visits

Round 2

Reviewer 1 Report

The revised version improved the manuscript.

However, I still have major concerns about the statistical work with comparisons that obviously show no statistical differences being reported as statistically significant. Please ask a statistician to check your results.

The ICER values make no sense, showing that they cannot be used to compare cost-effectiveness between regimens: in version 1, the authors included a % suppression of 85 for BIC/F/TAF, for a decrease in "ICER" of -38. In the revised version, the authors used another study for comparison with a % suppression of 94.8%. Nevertheless, they calculate a decrease in "ICER" of -175. So, based on the data presented, improving viral suppression by 10% caused further loss of cost-effectiveness (by 4-5 times!). This makes no sense, and the authors should have noted that. ICER is not the proper metric in this case, and it must be removed from the paper.

Author Response

We are grateful for these valuable comments and suggestions. Point-by-point responses are given below.

Reviwer 1:

The revised version improved the manuscript.

However, I still have major concerns about the statistical work with comparisons that obviously show no statistical differences being reported as statistically significant. Please ask a statistician to check your results.

The ICER values make no sense, showing that they cannot be used to compare cost-effectiveness between regimens: in version 1, the authors included a % suppression of 85 for BIC/F/TAF, for a decrease in "ICER" of -38. In the revised version, the authors used another study for comparison with a % suppression of 94.8%. Nevertheless, they calculate a decrease in "ICER" of -175. So, based on the data presented, improving viral suppression by 10% caused further loss of cost-effectiveness (by 4-5 times!). This makes no sense, and the authors should have noted that. ICER is not the proper metric in this case, and it must be removed from the paper.

Response: The aim of presenting ICER values is to represent the degree to which a strategy is more or less cost-effective than the reference approach. Although the method used to compare ICER values is always the same (dividing the difference in costs by the increase in effectiveness), correct interpretation requires knowledge of the methodology employed. Depending on the value (+ or -) of the cost increase and value (+ or -) of the increased benefit, a value with a "-" sign can be more cost-effective than one with a "+" sign in certain situations. This is well explained by Paulden M [1] using QALYs rather than effectiveness measures in his examples

In the present study, EICER values are compared between similar strategies, and none predominates over the others, as subsequently confirmed by our results. For this reason, we never suggest in the article that RPV/DTG excludes other approaches for being more cost-effective but rather indicate its relative position with respect to other strategies. This is now emphasized in the new revision. We are completely transparent in the article about the methods of analysis selected and about the data used as inputs (prices and effectiveness), allowing readers to make a critical evaluation of our results.

Our results only serve for comparison with the reference strategy and never for comparison between strategies, and we have not proposed any type of ranking among the different options. Hence, the negative or positive sign of a result should be interpreted in each case as the difference in cost-effectiveness with the reference approach. These points are now clarified in Results and Discussion.  

  1. (Paulden, M. Calculating and Interpreting ICERs and Net Benefit. PharmacoEconomics 38, 785–807 (2020). https://doi.org/10.1007/s40273-020-00914-6)

Reviewer 3 Report

Authors do not use a paired sample test as is needed for this design.  Clearly they do not understand statistics.

"The Student’s t-test was used for the analysis of quantitative variables with 101 normal distribution and Mann-Whitney U test for those with non-normal distribution. 102 The Kolmogorov-Smirnov test was applied to check the normality of variable 103 distributions. The Wilcoxon test was used to compare quantitative variables. p<0.05 was 104 considered significant in all tests. SPSS20.0 was used for statistical analyses"

Author Response

We are grateful for these valuable comments and suggestions. Point-by-point responses are given below.

Reviewer 2:

Authors do not use a paired sample test as is needed for this design.  Clearly they do not understand statistics.

Response. These points have been clarified in our revision of this section, as follows:

Statistical analysis: In the descriptive analysis, frequencies with 95% CI were calculated for qualitative variables, means with standard deviation for normally distributed quantitative variables, and medians with IQR for non-normally distributed quantitative variables. The Kolmogorov-Smirnov test was applied to check the normality of variable distributions. The paired samples Wilcoxon test was used to compare quantitative variables. p<0.05 was considered significant in all tests. SPSS 20.0 was used for statistical analyses.

Thank you so much

Yours sincerely. 

Carmen
